# LIGHTWEIGHT IN-CONTEXT TUNING FOR MULTIMODAL UNIFIED MODELS

## ABSTRACT

In-context learning (ICL) involves reasoning from given contextual examples. As more modalities comes, this procedure is becoming more challenging as the interleaved input modalities convolutes the understanding process. This is exemplified by the observation that multimodal models often struggle to effectively extrapolate from contextual examples to perform ICL. To address these challenges, we introduce **M**ulti**M**odal **I**n-conte**X**t **T**uning (M$^2$IXT), a lightweight module to enhance the ICL capabilities of multimodal unified models. The proposed M$^2$IXT module perceives an expandable context window to incorporate various labeled examples of multiple modalities (*e.g.*, text, image, and coordinates). It can be prepended to various multimodal unified models (*e.g.*, OFA, Unival, LLaVA) of different architectures and trained via a mixed-tasks strategy to enable rapid few-shot adaption on multiple tasks and datasets. When tuned on as little as 50K multimodal data, M$^2$IXT can boost the few-shot ICL performance significantly (*e.g.*, 18% relative increase for OFA), and obtained state-of-the-art results across an array of tasks including visual question answering, image captioning, visual grounding, and visual entailment, while being considerably small in terms of model parameters (*e.g.*, $\sim$20$\times$ smaller than Flamingo or MMICL), highlighting the flexibility and effectiveness of M$^2$IXT as a multimodal in-context learner.

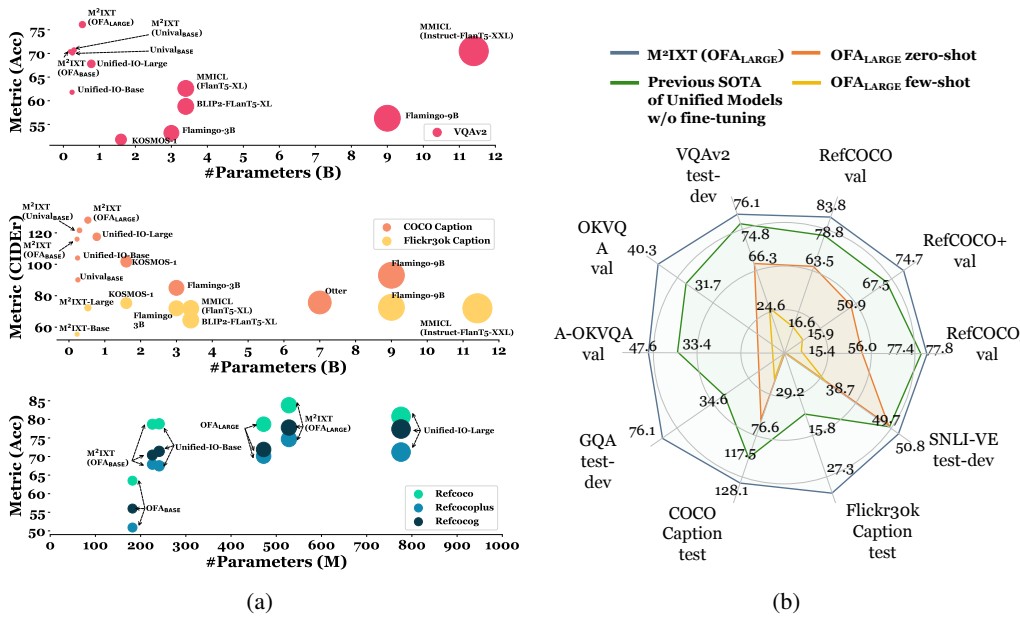

Figure 1: (a) M$^2$IXT surpasses existing multimodal models (*e.g.*, Flamingo, MMICL, Unified-IO) on multiple datasets and tasks while maintaining considerably small size; (b) The performance gain of M$^2$IXT (OFA$_{\text{LARGE}}$) over the base model (*i.e.*, OFA$_{\text{LARGE}}$) and previous state of the art is significant; Also, OFA$_{\text{LARGE}}$ cannot deal with in-context examples as evidenced by the comparison between its zero-shot and few-shot performances.

# 1 INTRODUCTION

In recent years, significant advancements have been made in the field of multimodal models (Radford et al., 2021; Wang et al., 2022d; Li et al., 2022b; 2023b; Wang et al., 2022b; Lu et al., 2023; Zhu et al., 2022b;a; Li et al., 2022a; Wang et al., 2022a; Alayrac et al., 2022), with vision-language models showcasing the most considerable improvements in performance and applicability. By jointly modeling diverse data modalities, multimodal models have set new benchmarks in various tasks, such as visual question answering (Antol et al., 2015; Marino et al., 2019; Schwenk et al., 2022; Hudson & Manning, 2019), visual grounding (Yu et al., 2016), and image captioning (Chen et al., 2015; Ordonez et al., 2011; Changpinyo et al., 2021; Plummer et al., 2015). A current trend in multimodal modeling (Chen et al., 2022a;b; Wang et al., 2022a; Lu et al., 2023) focuses on unifying different modalities and tasks through a sequence-to-sequence learning framework (Vaswani et al., 2017), aiming to build versatile models. These multimodal unified models are built on the principle of forgoing specifically designed modules, like detection heads in detectors (Ren et al., 2015) or segmentation heads in segmentors (Xiao et al., 2018), and instead incorporating all inputs and outputs within the same I/O space.

Despite their impressive generalization capabilities across multiple tasks and modalities, unified models often struggle to extrapolate from a few examples and perform few-shot learning on unseen datasets. Large language models (LLMs) (Brown et al., 2020; Liu et al., 2023b) have shown promising potential in few-shot adaptation through in-context learning (ICL) without updating their parameters. However, ICL has not been extensively explored in multimodal settings, where input sequences contain text, image, or other modalities and the integration of ICL capabilities into multimodal foundation models remains unclear and challenging.

The main challenges stem from the fact that during the pretraining phase, multimodal unified models are not adequately tailored for in-context learning, and the diversity of input modalities adds complexity to both the learning and inference processes, ultimately leading to suboptimal ICL performance. As evidenced by Figure 1 (b), the multimodal unified model, OFA (Wang et al., 2022a), fails to learn from contextual few-shot examples[1]. Specifically, adding few-shot examples to OFA even leads to a worse performance than its zero-shot inference, which is not an uncommon phenomenon (Alayrac et al., 2022; Awadalla et al., 2023; Tsimpoukelli et al., 2021). Potential reasons are: 1) the model encoder has never explicitly seen irregular modalities such as bounding box coordinates during pretraining; and 2) the added contextual examples can convolute the understanding of the test query. As such, it is necessary to have an additional module that can handle these in-context examples to fully harness the potential of ICL in multimodal settings, ultimately enabling them to reason more effectively from contextual few-shot examples.

In light of these challenges, we propose a **M**ulti**M**odal **I**n-conte**X**t **T**uning ($M^2$IXT) method for multimodal unified models. Recent work has demonstrated that, if trained appropriately, language models can be endowed with better ICL capability (Chen et al., 2022c; Min et al., 2022; Akyürek et al., 2022). Drawing inspiration from these findings, we design the $M^2$IXT module to encode in-context examples with multiple modalities and train it to perform in-context learning. $M^2$IXT is a lightweight module, and can be integrated into pretrained multimodal unified models (*e.g.*, OFA, LLaVA (Liu et al., 2023a), Unival (Shukor et al., 2023)) (model parameters are frozen) and trained for multiple tasks such as VQA, image captioning, and visual grounding by reusing a small portion of the original pretraining dataset of these multimodal unified models. In doing so, the $M^2$IXT module can be easily tuned with minimal computational overhead, and it learns to align the contextual examples and the test query of interest to make more accurate predictions, enabling a fast adaptation to downstream dataset. In summary, the contributions of our paper are as follows:

- We propose $M^2$IXT, an in-context tuning module explicitly designed to enable multimodal unified models to conduct in-context learning effectively. $M^2$IXT can deal with multimodal contextual examples and can be easily trained with a multi-task strategy.

- Empirical evaluations reveal that $M^2$IXT significantly improves the few-shot learning capabilities of existing multimodal unified models across diverse tasks and datasets, setting new performance benchmarks. Its strong performance in open-set evaluations underscores its potential as a versatile tool for a wide array of multimodal learning scenarios.

---

[1]We prepend multimodal in-context examples to queries.

- M$^2$IXT is lightweight and exhibits remarkable adaptability. As a plug-and-play module, M$^2$IXT can be easily integrated into multimodal unified models with different architectures without incurring much extra computational/memory overhead given its small model size.

## 2 RELATED WORK

**Vision Language Models (VLM).** It has been of long-standing interest to researchers to pretrain vision and language models to accomplish tasks such as visual question answering (Antol et al., 2015; Marino et al., 2019; Schwenk et al., 2022), visual grounding (Yu et al., 2016), captioning (Chen et al., 2015), and cross-modal retrieval (Lin et al., 2014). In recent times, there has been a significant growth in the development of foundation VLMs. These models are pretrained on a large scale and have proven to be effective in scaling up for modality encoding and ultimately improving the overall performance of downstream tasks. A typical combined model comprises modality-specific modules, *i.e.*, a vision module and a language module, which are connected via dual-encoder (Radford et al., 2021; Jia et al., 2021; Yuan et al., 2021) or mixture-of-experts structures (Wang et al., 2022b; Bao et al., 2022; Zhu et al., 2022a). They are also pretrained using different objectives, like image-text contrastive loss (Radford et al., 2021; Jia et al., 2021; Yuan et al., 2021; Bao et al., 2022; Yu et al., 2022), masked data modeling (Wang et al., 2022b; Bao et al., 2022), and maximum likelihood estimation (Zhu et al., 2022b;a; Li et al., 2022a; Wang et al., 2022a; Lu et al., 2023; Chen et al., 2022b). Several tasks related to vision-and-language are also being incorporated, starting with image-text matching (Radford et al., 2021; Jia et al., 2021) and gradually expanding to include more vision and language tasks (Wang et al., 2022b; Li et al., 2022a; Yu et al., 2022).

**Multimodal Unified Models.** Recently, there has been a trend to build unified models that handle multiple tasks and modalities with a sequence-to-sequence framework to mitigate the need for task-specific designs. Pix2seq (Chen et al., 2022a) and Pix2seq2 (Chen et al., 2022b) made an initial effort to combine object detection, segmentation, and keypoint detection into a single model by using a sequence-to-sequence architecture. Since then, unified models, with the ability to handle more tasks by representing data of various modalities in a unified I/O space (Wang et al., 2022a; Lu et al., 2023), have gained popularity and have become more prevalent. Most recently, Unival (Shukor et al., 2023) improves OFA by embedding video, image, text and audio modalities together and aligns them with transformers. In contrast to the encoder-decoder architecture, Uni-Perceivers (Zhu et al., 2022b;a; Li et al., 2022a) employ a transformer encoder-only architecture to align the likelihood of the predicted and target sequence. However, they are limited in the ability to facilitate generative tasks. Similarly, Painter (Wang et al., 2022c) employs a vision encoder but is restricted to dense labeling tasks that rely solely on image data. During the era of large models, there has been a trend to incorporate visual information into LLMs. LLaVA (Liu et al., 2023a), for example, injects vision transformer to LLaMA (Touvron et al., 2023) and achieves state-of-the-art accuracy on the ScienceQA benchmark.

**In-context Learning.** In-context learning (ICL, also known as few-shot prompting), popularized by GPT-3 (Brown et al., 2020), enables large language models to perform tasks by including a few input-output examples in the model's context (input) as a preamble, without updating any models parameters. ICL has been widely studied as an emergent capability of LLM (Wei et al.), but its application to the multimodal vision-language domain has only recently begun to be explored. Raw pretrained models, whether language or vision-language models, are not explicitly designed for in-context few-shot prompting during pretraining. An effective approach to enhancing the ICL capabilities of pretrained models is to fine-tune them by prepending a few labeled in-context examples to the target input. For example, Chen et al. (Chen et al., 2022c) propose an in-context tuning method that meta-trains an LM to learn to adapt to new tasks from a few examples. More relevant to our work is Flamingo (Alayrac et al., 2022), which is trained to endow VLM with in-context few-shot learning capabilities. Flamingo takes interleaved visual data and text as input and generates free-form text as output, and uses LLM as the backbone. Most recently, MMICL (Zhao et al., 2023) and Otter (Li et al., 2023a) proposed to finetune the large vision-language model via large-scale in-context learning. However, they show marginal performance gains with substantial training cost while our method is more efficient and lightweight. Furthermore, our primary focus lies in enhancing the ICL capability of multimodal unified models.

Figure 2: The architecture of the proposed M$^2$IXT. It incorporates multimodal contextual examples as input and can be integrated into multimodal unified models with varipus archotectures.

## 3 THE PROPOSED METHOD: M$^2$IXT

To endow multimodal unified models with the ability to perform in-context few-shot reasoning, we propose multimodal in-context tuning (M$^2$IXT). Specifically, the M$^2$IXT module takes as input a few multimodal labeled examples. Each contextual example consists of an image, a text instruction, and the corresponding answer. The M$^2$IXT module is compatible with multiple tasks, including visual question answering, image captioning, visual grounding, *etc.*, and can be prepended to multimodal unified models of different architectures.

### 3.1 THE ARCHITECTURE OF M$^2$IXT

Following previous practices in multimodal unified models, an encoder-decoder transformer framework (*e.g.*, OFA (Wang et al., 2022a), Unival (Shukor et al., 2023)), or decoder-only transformer framework (*e.g.*, LLaVA (Liu et al., 2023a)) can be adopted as the backbone. These miltimodal unified models generate target sequences conditioned on the input source sequences, and are usually optimized by minimizing the negative log-likelihood loss, $\ell = -\sum_{i=1}^{|y|} \log P_\theta(y_i|\hat{y}_{1:i-1}, x, s)$, where $\theta$ is the model parameters, $x$ is the input image, $s$ is the instruction, and $\hat{y}_{1:i-1}$ is the $i-1$ preceding tokens of output $y$.

In an ICL setting, suppose we have $N$ contextual examples of [Image, Instruction, Target] triples, and the $i^{\text{th}}$ example is denoted as $C_i$. Contextual examples are separated by by adding <bos> and <eos> to the beginning and end of each example. The M$^2$IXT module takes the $N$ multimodal examples as input and outputs a sequence of token embeddings which can be concatenated with the query sequence embeddings. To handle multimodal examples, the M$^2$IXT module comprises three tunable components: a visual encoder (*e.g.*, ResNet or ViT), a text embedding dictionary, and a target embedding network. The target embedding network can process conventional modalities (*e.g.*, text tokens) as well as special modalities such as bounding box coordinate tokens. M$^2$IXT is lighweight as it only brings 40M~60M additional tunable parameters. Figure 2 illustrates how the M$^2$IXT module is integrated into a multimodal unified model. The M$^2$IXT module is decoupled as a standalone module by freezing the original multimodal unified model, which minimizes the training overhead and accelerates adaptations.

M$^2$IXT samples tokens based on the model likelihood $P(y_i|\hat{y}_{1:i-1}, x, s, C_1, ..., C_N)$ conditioned on in-context sequences $[C_1, ..., C_N]$. The M$^2$IXT training objective function is the same as the one used for multimodal unified model apart from the added additional input contextual examples,

$$\ell = -\sum_{i=1}^{|y|} \log P_\theta(y_i|\hat{y}_{1:i-1}, x, s, C_1, ..., C_N), \tag{1}$$

where, $x$, $s$ and $y$ represents the query image, query task instruction, and query target ground truth, respectively. Standard optimization methods, such as causal masks and teacher forcing, are leveraged in the training process. In addition, we adopt random resize, center crop, RandAug, horizontal flip, large Scale Jittering (Ghiasi et al., 2021) for image data augmentation.

To perform ICL during inference, we need to draw labeled examples as the context for each test query. To this end, we randomly sample in-context examples from the evaluation set if labels are available following the same setting as Flamingo (Alayrac et al., 2022) and Painter (Wang et al., 2022c); If labels are not accessible from the evaluation set (*e.g.*, test split or online evaluations), we

draw samples from any arbitrary public datasets (Antol et al., 2015; Chen et al., 2015), allowing for a better generality. Additionally, beam search is adopted to ensure generation quality.

## 3.2 TRAINING OF M²IXT

We detail the training procedure of M²IXT by taking OFA as the backbone for brevity (We adopt the same setting for Unival.). We integrate M²IXT into different variants (*e.g.*, OFA$_{BASE}$ and OFA$_{LARGE}$) of OFA. The text embedding dictionary and the target embedding network are initialized using the pretrained embeddings of OFA, while the visual input is embedded using a ResNet (*i.e.*, ResNet-101 or ResNet-152)[2]. Similarly, for LLaVA, we freeze all its parameters and only tune the M²IXT module.

### 3.2.1 UNIFIED DATA FORMAT

Multimodal learning involves unifying language and image data through a tokenizer and an embedding network that projects them into discrete tokens represented as vectors in hidden dimensions. These tokens are then serialized into sequences for each sample. While the ordering of tokens may vary, most methods follow a serialization of [`Image, Instruction or Command, Target`], which is separated into a source sequence of [`Image, Instruction`] and a target sequence of [`Target`] during implementation. For multi-tasking, the instruction or command varies for different tasks, such as *"Detect the objects"* for object detection and *"What does the image describe?"* for image captioning. This allows the unified model to generate output based on the input. Text modality tokenization (Sennrich et al., 2016) is initialized using a linguistic vocabulary, and pretrained visual network such as ResNet (He et al., 2016) is used for image modality tokenization and embedding. In some cases, a separate set of vocabulary is created to represent special data in `Target` like coordinates (*e.g.*, bounding box coordinates) such as "`<bin>+coordinate`" (*e.g.*, "`<bin>456`") to differentiate them from regular numbers.

### 3.2.2 MIXED-TASKS TRAINING

The M²IXT module is trained with a unified dataset that contains multiple tasks such as image captioning, visual question answering, visual grounding, *etc.*. We employ a task-heterogeneous batches strategy following (Aghajanyan et al., 2021) by shuffling all the samples randomly, with which each batch contains multimodal in-context examples of different tasks. This encourages multiple tasks to learn a shared representation, enabling an easier transfer to unseen data (Xie et al., 2019; Marino et al., 2019; Schwenk et al., 2022; Plummer et al., 2015).

We lay out the pretraining tasks and datasets in details. It is worth noting that all these datasets are sampled from the OFA pretraining dataset. In specific, we adopt several vision and language tasks and datasets for M²IXT training, including visual question answering (VQAv2 (Antol et al., 2015)), image captioning (COCO (Chen et al., 2015), SBU (Ordonez et al., 2011), CC12M (Changpinyo et al., 2021)), visual grounding (RefCOCO, RefCOCO+, and RefCOCOg (Yu et al., 2016)), masked image modeling (ImageNet-21k (Deng et al., 2009)), as well as object detection task (OpenImage (Kuznetsova et al., 2020)). By default, we randomly sample part of the mentioned vision and language data and randomly select 25,000 samples from both masked image modeling (ImageNet-21k) and object detection (OpenImage), resulting in only 0.5M samples which are $\sim 50\times$ less than the original OFA pretraining dataset. Section 4.4 examines the impact on the model's performance by varying the sampling percentage. We transform all the images, instructions, and targets in an in-context manner and randomly sample them from the dataset to construct the in-context examples.

## 4 EXPERIMENTS

### 4.1 EXPERIMENTAL SETUP

We randomly select in-context examples from the mixed-tasks dataset, and we use random ordering for the in-context samples in single sequence. We set pretrained image size as $384 \times 384$, and use the visual encoder to divide it into $24 \times 24$ patches, resulting in 576 tokens for one image. Together with

---

[2]Empirically, we found that it obtains comparable performance with ViT (CLIP ViT-B).

Table 1: Few-shot experiments of M$^2$IXT (OFA): Multi-tasking evaluation on VQAv2, COCO Caption, and SNLI Visual Entailment.

| Methods | VQAv2 | | COCO Caption test | | | | SNLI-VE |
|---|---|---|---|---|---|---|---|
| | val | test-dev | BLEU@4 | METEOR | CIDEr | SPICE | dev |
| OFA$_{BASE}$ | 66.3 | 69.7 | 21.7 | 20.8 | 76.6 | 16.1 | 49.7 |
| M$^2$IXT (OFA$_{BASE}$) | 70.1 | 70.4 | 34.6 | 28.3 | 116.0 | 21.8 | 50.7 |
| OFA$_{LARGE}$ | 73.0 | 74.8 | 22.2 | 20.4 | 75.0 | 15.3 | 41.0 |
| M$^2$IXT (OFA$_{LARGE}$) | 75.7 | 76.1 | 37.8 | 30.2 | 128.1 | 23.0 | 42.6 |

Table 2: Few-shot experiments of M$^2$IXT (OFA): Multi-tasking evaluation on visual grounding task with RefCOCO/RefCOCO+/RefCOCOg dataset.

| Methods | RefCOCO | | | RefCOCO+ | | | RefCOCOg | |
|---|---|---|---|---|---|---|---|---|
| | val | testA | testB | val | testA | testB | val | test |
| OFA$_{BASE}$ | 63.5 | 67.3 | 60.4 | 50.9 | 56.8 | 44.7 | 56.0 | 56.2 |
| M$^2$IXT (OFA$_{BASE}$) | 78.7 | 83.8 | 72.1 | 67.9 | 76.2 | 57.4 | 70.4 | 71.6 |
| OFA$_{LARGE}$ | 78.7 | 82.6 | 75.2 | 70.1 | 77.0 | 63.8 | 71.9 | 72.2 |
| M$^2$IXT (OFA$_{LARGE}$) | 83.8 | 88.2 | 78.3 | 74.7 | 82.8 | 64.9 | 77.8 | 78.1 |

Table 3: Comparison experiments with SOTA unified models: all models are under multi-tasking evaluation on VQA, image captioning, and visual grounding tasks w/o tuning.

| Methods | #Params. | VQAv2 test-dev | COCO Caption test | | RefCOCO/RefCOCO+/RefCOCOg val |
|---|---|---|---|---|---|
| | | acc | B@4 | CIDEr | acc |
| Uni-Perceiver-MoE$_{BASE}$ | 167M | - | 33.6 | - | - |
| Uni-Perceiver-v2$_{BASE}$ | 308M | - | - | 116.9 | - |
| Flamingo-3B | 3B | 53.2 | - | 85.0 | - |
| Unified-IO$_{SMALL}$ | 71M | 57.7 | - | 80.1 | 58.5/44.7/53.3 |
| Unified-IO$_{BASE}$ | 241M | 61.8 | - | 104.0 | 78.8/67.5/71.4 |
| Unified-IO$_{LARGE}$ | 776M | 67.8 | - | 117.5 | 80.8/71.2/77.4 |
| Unival$_{BASE}$ | 250M | 70.1 | - | 90.1 | -/70.8/- |
| Otter | 7B | - | - | 75.7 | - |
| MMICL (Flan-T5-XL) | 3.4B | 62.6 | - | - | - |
| MMICL (Flan-T5-XXL) | 11.4B | 70.5 | - | - | - |
| M$^2$IXT (OFA$_{BASE}$) | 226M | 70.4 | 34.6 | 116.0 | 78.7/67.9/70.4 |
| M$^2$IXT (Unival$_{BASE}$) | 294M | 70.7 | - | 121.6 | -/72.0/- |
| M$^2$IXT (OFA$_{LARGE}$) | 528M | **76.1** | **37.8** | **128.1** | **83.8/74.7/77.8** |

the instructions and label tokens, M$^2$IXT is learned to handle a large context window of ~3k tokens on average. We use Adam optimizer for model learning, and we set the maximum epoch number to 20, weight decay to 0.01, warmup ratio to 0.01, and the initial learning rate to $10^{-4}$ with a cosine scheduler. Based on empirical evidence, it takes around 3 days to train M$^2$IXT on a machine with 16 NVIDIA Tesla V100-16GB GPUs, using a pretraining data setting of 0.5M. However, with 50K pretraining data setting (0.2% of OFA data, as shown in Figure 6), it can be finished in approximately 7 hours.

## 4.2 PERFORMANCE BOOST WITH M$^2$IXT

**Integrating M$^2$IXT into OFA.** We first demonstrate how M$^2$IXT can enhance the performance of the popular multimodal unified model, *i.e.*, OFA, under a few-shot setting, using 2-shots by default. As illustrated in Table 1 and Table 2, outfitting OFA with M$^2$IXT can substantially improve performance across multiple tasks and datasets, with an average 25% and 11.4% relative performance gain for OFA$_{BASE}$ and OFA$_{LARGE}$, respectively. The results of M$^2$IXT (OFA$_{TINY}$ and M$^2$IXT (OFA$_{SMALL}$) are provided in the appendix A.2. These results affirm the effectiveness and adaptability of M$^2$IXT when incorporated into backbones of varying model sizes. Meanwhile, when conducting a few-shot on OFA directly, it exhibits poorer results (Figure 1) (b), suggesting M$^2$IXT with early exposure to contextual examples can significantly enhance its performance. A similar performance boost in unimodal model has been reported in (Chen et al., 2022c).

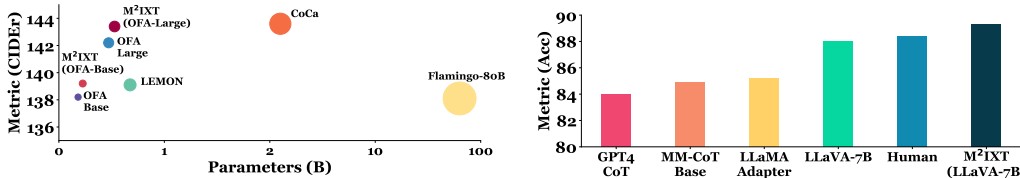

Figure 3: Left: Full fine-tuning on COCO Caption in cross-entropy optimization. Circle size relatively denote the overall parameters of each model; Right: results of M²IXT (LLaVA-7B) on the ScienceQA-`test` dataset.

Table 4: Unseen task Experiments: models are evaluated on A-/OKVQA, Flickr30k w/o tuning.

| Methods | #Params. | OKVQA val | A-OKVQA val | Flickr30k BLEU@4 | Flickr30k CIDEr |
|---|---|---|---|---|---|
| Frozen (Tsimpoukelli et al., 2021) | 7.1B | 5.9 | - | - | - |
| Few-VLM (Jin et al., 2022) | 785M | 16.5 | - | - | - |
| Uni-Perceiver-MoE-L (Zhu et al., 2022a) | 505M | - | - | 15.8 | - |
| VLKD (Dai et al., 2022) | 832M | 13.3 | - | - | - |
| BLIP-2 ViT-g OPT2.7B (Li et al., 2023b) | 3.8B | 31.7 | - | - | - |
| MMICL (FlanT5-XL) (Zhao et al., 2023) | 3.4B | - | - | - | 71.9 |
| MMICL (Instruct-FlanT5-XXL) (Zhao et al., 2023) | 11.4B | - | - | - | 72.0 |
| M²IXT (OFA$_{BASE}$) | 226M | 34.3 | 40.7 | 21.9 | 55.5 |
| M²IXT (OFA$_{LARGE}$) | 528M | **40.3** | **47.6** | **27.3** | **72.3** |

**Integrated with other Multimodal Models.** We apply M²IXT to decoder-only LLaVA-7B (Liu et al., 2023a) and encoder-decoder Unival (Shukor et al., 2023) in Figure 3 and Table 3. Our M²IXT modules are constructed following the method in our paper and appended to LLaVA and Unival for in-context tuning. For LLaVA-7B, we employ pretrained weights from the ScienceQA dataset for initialization, whereas for Unival, we use its stage2 pretrained weights for initialization. It's important to emphasize that only the M²IXT modules are open to training, while all other parameters remain fixed. As depicted in Figure 3(Right) and Table 3, M²IXT delivers significant enhancements across all datasets for both models. This underscores the remarkable adaptability of M²IXT.

**Comparison with Previous SOTA on Few-shot Learning.** There are only a handful of multimodal unified models that evaluate their few-shot/zero-shot learning capabilities on public benchmarks. Here we compare M²IXT with Uni-Perceiver-MoE (Zhu et al., 2022a), Uni-Perceiver-v2 (Li et al., 2022a), Unified-IO (Lu et al., 2023), and Flamingo (Alayrac et al., 2022) without any further fine-tuning. From Table 3, we make the following observations. (1) With M²IXT, we obtain the state-of-art performance on almost all datasets, and compared with the best baselines, the improvement is considerably substantial; (2) While being smaller in model size, it still exhibits comparable results to counterparts (*e.g.*, Unified-IO$_{LARGE}$) which are $\sim 3\times$ to $\sim 10\times$ larger; (3) Although trained to handle few-shot examples, Flamingo, MMICL, and Otter (4-shots) with billions of parameters underperforms other methods, which underscores the superiority of M²IXT as a multimodal in-context learner.

**Full Fine-tuning.** As an additional module, M²IXT will not degrade the full fine-tuning performance of the backbone unified models. We unfreeze all the model parameters and perform full finetuning on COCO Caption by simply replacing the mixed-tasks datasets with the COCO Caption training set. As shown in Figure 3, our method achieves a good overall result over baselines, with 139.2 CIDEr on OFA$_{BASE}$ + M²IXT and 143.4 CIDEr on OFA$_{LARGE}$ + M²IXT, beating in-context counterpart 80B Flamingo, generalized decoding model X-Decoder, and on par with multimodal large models CoCa. More results are listed in appendix A.2.

## 4.3 OPEN SET EVALUATION

Zero-shot inference on unseen datasets is becoming a crucial benchmark for multimodal models (Zhu et al., 2022b;a; Li et al., 2023b; Radford et al., 2021). Meanwhile, ICL shows promising results on unseen dataset (Marino et al., 2019; Schwenk et al., 2022; Plummer et al., 2015; Hudson & Manning, 2019). Thus, we tested the efficacy of M²IXT in an open-set evaluation, using three datasets: OKVQA (Marino et al., 2019), A-OKVQA (Schwenk et al., 2022), and Flickr30k (Plummer et al., 2015). These datasets contain complex questions that often require external knowledge to answer accurately. We compared M²IXT with several baselines, including in-context multimodal

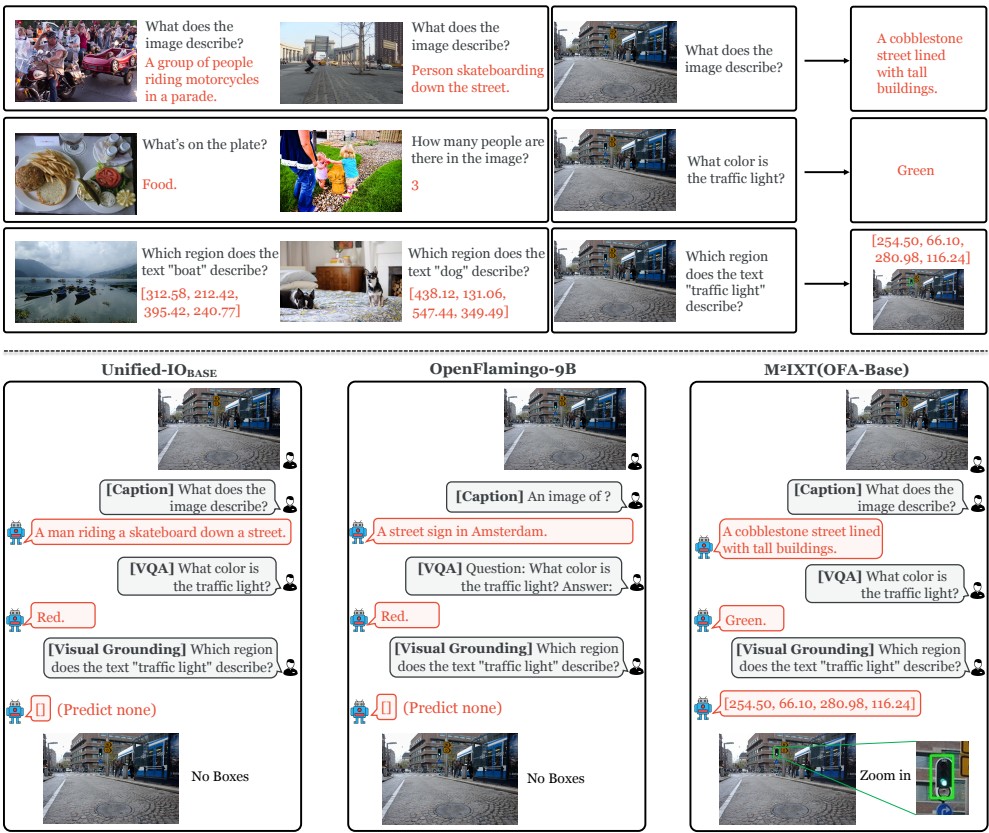

Figure 4: M$^2$IXT Visualizations for multimodal tasks. Inputs select tasks and input query text, M$^2$IXT respond and give right answers. The in-context examples are from public dataset. Upper shows the few-shot inference demo of M$^2$IXT across VQA, image captioning, and visual grounding tasks, lower shows the comparison between SOTA unified model Unified-IO$_{\text{BASE}}$ and in-context counterpart OpenFlamingo-9B(Awadalla et al., 2023).

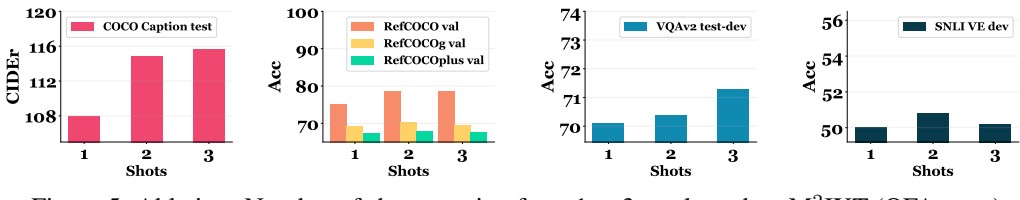

Figure 5: Ablation: Number of shots ranging from 1 to 3, evaluated on M$^2$IXT (OFA$_{\text{BASE}}$).

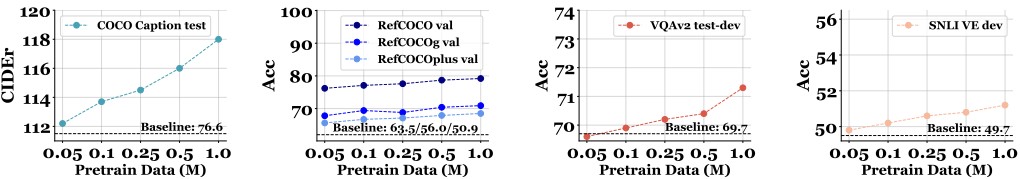

Figure 6: Ablations: Number of pretraining data samples, unit million (M) is used for measurement.

Frozen (Tsimpoukelli et al., 2021), prompt tuning Few-VLM (Jin et al., 2022), distilled VLCLIP (Dai et al., 2022), zero-shot unified model Uni-Perceiver-MoE (Zhu et al., 2022a), and multimodal LLM-based BLIP-2 (Li et al., 2023b). The empirical results, as shown in Table 4, suggest that M$^2$IXT is highly effective in leveraging external knowledge and reasoning from diverse multimodal examples in an ICL setting.

### 4.4 MODEL ANALYSIS

We conduct thorough model analyses on M$^2$IXT with OFA as the backbone model.

**Case Study.** For an intuitive understanding of M$^2$IXT, we showcase a few real examples to illustrate the in-context learning process. The upper three cases in Figure 4 demonstrate that M$^2$IXT can effectively handle diverse inputs in a row following the in-context learning pattern in a multimodal setting. Also, we compare M$^2$IXT (OFA$_{\text{BASE}}$) with Unified-IO and Flamingo (*i.e.*, OpenFlamingo-9B(Awadalla et al., 2023)) on performing multi-tasks. We use 2-shots for OpenFlamingo-9B, same as our 2-shots M$^2$IXT. The comparison is also in Figure 4, from which we observe that M$^2$IXT (OFA$_{\text{BASE}}$) can give more reasonable responses for captioning and correctly answered the challenging VQA and Visual grounding questions, while Unified-IO and Flamingo either deliver wrong responses or invalid outputs. Moreover, we notice that M$^2$IXT (OFA$_{\text{BASE}}$) exhibits a noteworthy capability in precisely localizing small objects, *e.g.*, the traffic light in Figure 4 occupies 26×50 pixels.

**Ablation Study on Number of Shots.** In our evaluations, we kept the number of in-context examples constant for simplicity. However, it's reasonable to question if the number of in-context examples can significantly affect performance. To investigate this, we vary the number of in-context examples from 1 to 3 for several tasks and report the results in Figure 5. We observe that increasing the number of examples from 1 to 2 can offer more performance boost while further adding it to 3 can only bring marginal benefits. One possible explanation is that, unlike natural language prompts, each multimodal in-context example requires a considerably large token length, which may aggravate the difficulty of inference. Therefore, we recommend setting the number within the range of 1-3 as it strikes a balance between resource utilization and accuracy.

**Ablation Study on Size of the Mixed-Tasks Training Set.** In this section, we explore if M$^2$IXT could enhance the performance of multimodal models with less amount of data. Figure 6 presents our findings on how the size of mixed-tasks training set affects the model performance. Surprisingly, using 50K (0.2% of the OFA data) pretraining data achieves quite decent performance for most tasks, compared with OFA baselines in Figure 6. Specifically, it only takes a few hours to train an M$^2$IXT (OFA$_{\text{BASE}}$) model with 50K pretraining data. Also, M$^2$IXT shows quite impressive scaling-up ability when the data percentage increases, indicating that the few-shot reasoning ability can be further enhanced via larger-scale training.

**Ablation Study on Tasks of the Training Set.** We conduct an ablation experiment by removing each type of task and then retraining M$^2$IXT. Results in Figure 7 illustrate that when we eliminate tasks that exhibit high relevance, there is a noticeable performance drop. For instance, ablating the visual grounding task deteriorates the performance significantly. Interestingly, the detection and MIM tasks do not contribute to improving the downstream caption task. Nevertheless, we retain them for their positive impact on overall performance.

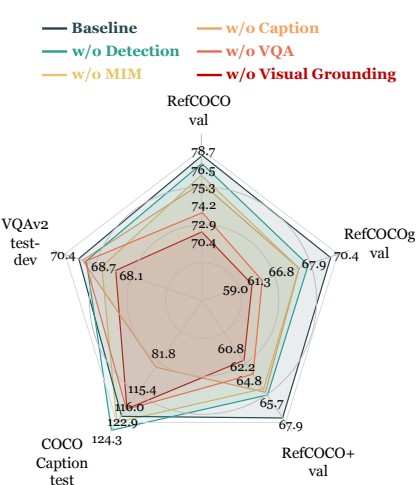

Figure 7: Results of ablating training tasks, evaluated on M$^2$IXT (OFA$_{\text{BASE}}$).

## 5 CONCLUSION

We propose a lightweight multimodal in-context tuning method, M$^2$IXT for multimodal unified models, endowing them with the reasoning ability to infer from in-context samples. With M$^2$IXT, we can quickly adapt unified models to unseen datasets and an open-set world with minimal computational overhead. Empirically evaluations show that M$^2$IXT can effectively boost the few-shot learning performance of existing multimodal unified models and obtain state-of-the-art results on multiple datasets and tasks. We hope that M$^2$IXT will spur further research on bolstering the multimodal ICL capabilities to improve the usability and accessibility of multimodal unified models.

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

## A  APPENDIX

### A.1  IMPLEMENTATION DETAILS

#### A.1.1  ARCHITECTURE

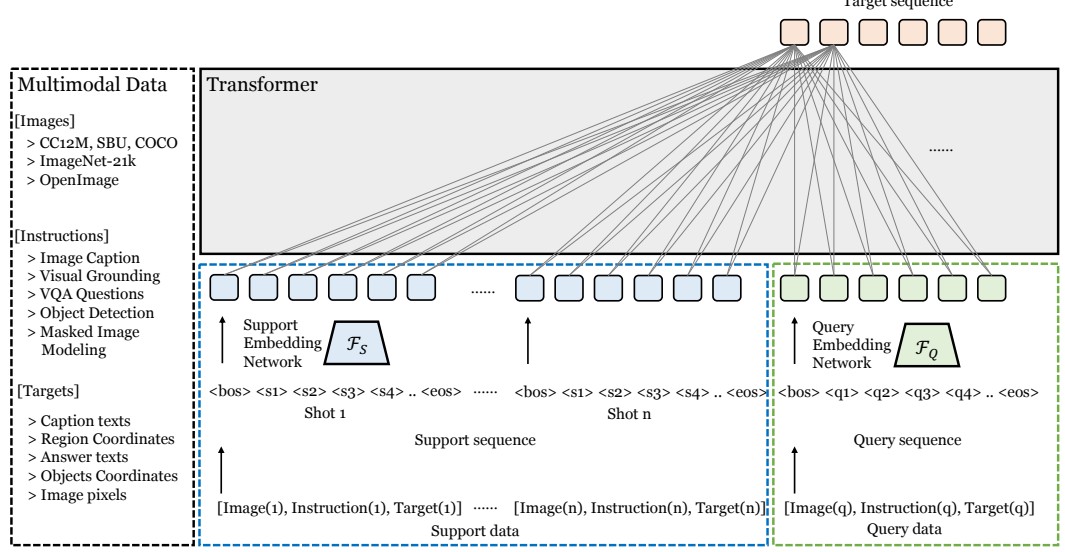

Figure 8: Detailed framework diagram.

To maintain consistency with previous multimodal unified models, a framework based on the transformer architecture (Vaswani et al., 2017) is used for all pretraining, finetuning, and evaluation tasks. Take encoder-decoder backbone as an example, it is composed of consecutive transformer layers, where each encoder layer has a self-attention and a feed-forward network (FFN), and each decoder layer has a self-attention, FFN, and cross-attention for linking the decoder to the encoder output representations. $M^2$IXT (OFA) ranges from tiny to large, as illustrated in Figure 5. As for decoder-only backbone, it is composed of self-attention transformer layers, and we apply causal masks in training. When OFA is utilized as backbone, the number of parameters grows from 56M to 528M as the vision embedding, hidden size, Multi-Attention Heads and encoder/decoder layers scale up.

Table 5: M$^2$IXT Architecture details. MA Heads, Enc. and Dec. represents Number of Multi-Attention Heads, transformer encoder layers and transformer decoder layers.

| Model | #Params.(Trainable) | Vision Embedding | Hidden Size | MA Heads | Enc. | Dec. |
|---|---|---|---|---|---|---|
| M$^2$IXT (OFA$_{TINY}$) | 56M(26M) | ResNet50 | 256 | 4 | 4 | 4 |
| M$^2$IXT (OFA$_{MEDIUM}$) | 137M(44M) | ResNet101 | 512 | 8 | 4 | 4 |
| M$^2$IXT (OFA$_{BASE}$) | 226M(44M) | ResNet101 | 768 | 12 | 6 | 6 |
| M$^2$IXT (OFA$_{LARGE}$) | 528M(56M) | ResNet152 | 1024 | 16 | 12 | 12 |

Table 6: Experiments w/o tuning: Multi-tasking evaluation on VQAv2, COCO Caption, and SNLI Visual Entailment w.r.t.small models.

| Methods | VQAv2 | | COCO Caption test | | | | SNLI-VE |
|---|---|---|---|---|---|---|---|
| | val | test-dev | BLEU@4 | METEOR | CIDEr | SPICE | dev |
| OFA$_{TINY}$ | 47.9 | 50.7 | 19.6 | 18.1 | 65.1 | 13.7 | 36.8 |
| M$^2$IXT (OFA$_{TINY}$) | 58.5 | 59.6 | 29.3 | 24.6 | 93.4 | 18.1 | 43.5 |
| OFA$_{MEDIUM}$ | 62.5 | 64.5 | 16.9 | 18.1 | 62.8 | 14.1 | 45.6 |
| M$^2$IXT (OFA$_{MEDIUM}$) | 65.9 | 66.9 | 34.4 | 27.9 | 113.7 | 21.1 | 46.1 |

### A.1.2 EXPERIMENTAL DETAILS

**Dataset Build.** In joint training for mixed-tasks, we build mixed-dataset from several public datasets. In main paper, we mention that we select data from SBU (Ordonez et al., 2011), COCO (Chen et al., 2015), CC12M (Changpinyo et al., 2021), RefCOCO/+/g (Yu et al., 2016) and VQAv2 (Antol et al., 2015). In detail, we select 50k samples from SBU, COCO, CC12M, RefCOCO, RefCOCO+ and RefCOCOg, to balance between three tasks (VQA, visual grounding, captioning), we select 150k samples from VQAv2, which results to 0.45M pre-training samples. Added with masked image modeling task (ImageNet-21k 25k samples) and object detection task (OpenImage 25k samples), we get 0.5M samples in total for mixed-dataset. In our ablation study about size of mixed-dataset, for 0.05M, 0.1M and 0.25M setting, we simply scale down them by respective ratios ($\times 0.1$, $\times 0.2$, $\times 0.5$), for 1M setting, we also scale up them by ratios ($\times 2$) except we select less samples for those datasets which do not have adequate images (*e.g.*, RefCOCOg do not have 100k samples), so 1M setting owns less than 1M samples. For batch collation, we pad each in-context example in same shot position (*e.g.*, the $1^{st}$ shot or the $2^{nd}$ shot) to the same size in each mini-batch, which makes it easy for training parallelization. We also follow OFA (Wang et al., 2022a) to create data samples (*e.g.*, transforming the caption sample into VQA sample, `"Question: Does the image describe {this caption}? Answer: Yes/No"`) for in-context tuning.

**Training Details.** In pretraining, we train all M$^2$IXT (OFA) in 20 epochs, with initial learning rate 1e-4, batch size 1 per GPU, cosine learning rate decay with warmup ratio 0.01, weight decay 0.01, gradient clip norm 5.0. In transformer model, we set the encoder and decoder drop path rate as 0.1, the dropout probability 0.1. In data loading, we set max token length 80 and 30 for source sequence and target sequence respectively, and we set the coordinate vocabulary's size as 1000, that is, the float coordinates are quantized into 1000 bins. In finetuning, we train the models following OFA's setting, with transformer parameters activated. We tune the visual grounding, VQA, and SNLI-VE task for 10, 15, and 5 epochs, respectively. We set the batch size as 1 and gradient accumulation as 8 for matching with OFA's. We also rerun the OFA baseline in this finetuning setting for fair comparison.

### A.2 EXTENDED EXPERIMENTS

**Small Models.** Apart from base and large models in the main text, we also provide multi-tasking evaluation for small models M$^2$IXT (OFA$_{TINY}$) and M$^2$IXT (OFA$_{MEDIUM}$). As illustrated in Table 6 and 7, on average, we outperform the baseline OFA$_{TINY}$ and OFA$_{MEDIUM}$ by relative **37%** and **15%** respectively in VQA, captioning, SNLI-VE and visual grounding tasks.

**Full Fine-tuning.** In addition to the fine-tuning for captioning task in main paper, we also conduct full-finetuning experiments for VQA, SNLI-VE and visual grounding tasks, as shown in Table 8 and 9. From performance we can surpass the baseline OFA by small margin, but still fall behind large models such as BEiT3 and CoCa due to lack of parameters. Due to lack of resources, we leave the

Table 7: Experiments w/o tuning: Multi-tasking evaluation on visual grounding task with Ref-COCO/RefCOCO+/RefCOCOg dataset w.r.t.small models.

| Methods | RefCOCO | | | RefCOCO+ | | | RefCOCOg | |
|---|---|---|---|---|---|---|---|---|
| | val | testA | testB | val | testA | testB | val | test |
| $OFA_{TINY}$ | 44.3 | 48.6 | 37.8 | 35.4 | 39.5 | 28.0 | 38.1 | 38.7 |
| $M^2IXT$ ($OFA_{TINY}$) | 63.9 | 69.4 | 57.1 | 52.0 | 59.4 | 42.0 | 56.0 | 56.5 |
| $OFA_{MEDIUM}$ | 67.4 | 72.2 | 61.5 | 55.0 | 62.6 | 46.2 | 59.6 | 60.5 |
| $M^2IXT$ ($OFA_{MEDIUM}$) | 74.1 | 80.0 | 67.3 | 62.7 | 70.9 | 51.1 | 65.8 | 66.3 |

Table 8: Comparison Experiments: Full fine-tuning on visual question answering w.r.t.VQAv2 and visual entailment w.r.t.SNLI-VE.

| Methods | #Params. | VQAv2 | SNLI-VE |
|---|---|---|---|
| | | test-dev | dev |
| SimVLM (Wang et al., 2022d) | - | 80.0 | 86.2 |
| Flamingo (Alayrac et al., 2022) | 80B | 82.0 | - |
| CoCa (Yu et al., 2022) | 2.1B | 82.3 | 87.0 |
| BEiT3 (Wang et al., 2022b) | 1.9B | 84.2 | - |
| $OFA_{BASE}$ (Wang et al., 2022a) | 182M | 78.0 | 89.3 |
| $OFA_{LARGE}$ (Wang et al., 2022a) | 472M | 80.3 | 90.3 |
| $M^2IXT$ ($OFA_{BASE}$) | 226M | 78.3 | 89.5 |
| $M^2IXT$ ($OFA_{LARGE}$) | 528M | 80.7 | 90.6 |

scaling up to future work, when we are able to increase the transformer encoder-decoder layers and replace the vision embedding with ViT-G (Dosovitskiy et al., 2021) to produce over 1B unified model, and train $M^2IXT$ with massive data.

**n-Pretrained/m-Evaluated shots matrix.** In previous scenarios, the training dataset only consists of one type of samples and model is tested under same-shot setting. For example, 2-shot training corresponds to 2-shot samples while it is also tested under 2-shot setting. To evaluate the effectiveness of n-shots models in m-shots evaluation settings, we conducted experiments using 1/2/3/4-shots settings. We created a performance matrix in Table 11 to assess the effect. The x-axis represents the evaluation setting, while the y-axis represents the pretrained setting. For instance, the $2^{nd}$ row and $3^{rd}$ column indicate the evaluation of a 2-shots pretrained model in a 3-shots setting. The results confirm that if the settings do not match, the performance will decrease. However, if models are trained with more shots, they can still perform well in evaluations with fewer shots (*e.g.*, a 3-shots pretrained model can achieve over 100 CIDEr in caption, even though it is less than 3-shots).

**Stable-shots Training.** We also conducted an experiment where we included both 1/2/3-shots samples together by uniformly sampling them in one iteration during training, which stabilizes our $M^2IXT$ and improve baseline by 3 points on average. As shown in Table 11, the randomly sampled shots for pretraining performed well in all evaluated shots, surpassing all previous results when trained on the same shots setting.

**Training and Inference Overhead.** We conduct experiments to test the training time of one training sample per GPU, see Table 12. With large context window, $M^2IXT$ brings training overhead against OFA. However, as is stated in paper, we can obtain decent performance in fast 1-epoch training, which takes 4 hours. And we also test the inference time (Table 13) for $M^2IXT$ (2 shots), using COCO Captioning task data.

## A.3 MORE OPEN-ENDED CASES

We provide more cases using $M^2IXT$, to test the generality for several vision and language tasks via user inputs. For test images, we randomly collect them in the Flickr [3]. For caption, the instruction

---
[3] https://www.flickr.com

Table 9: Comparison Experiments: Full fine-tuning on visual grounding task w.r.t.RefCOCO, Ref-COCO+ and RefCOCOg.

| Methods | #Params. | RefCOCO val | RefCOCO+ val | RefCOCOg val |
|---|---|---|---|---|
| UNITER (Chen et al., 2020) | - | 81.4 | 75.9 | 74.8 |
| VILLA (Gan et al., 2020) | 80B | 82.4 | 76.2 | 76.2 |
| MDETR (Kamath et al., 2021) | - | 86.8 | 79.5 | 81.6 |
| UNICORN (Yang et al., 2021) | 1.9B | 88.3 | 80.3 | 83.4 |
| OFA$_{\text{BASE}}$ (Wang et al., 2022a) | 182M | 86.3 | 80.2 | 81.2 |
| OFA$_{\text{LARGE}}$ (Wang et al., 2022a) | 472M | 89.7 | 84.7 | 85.6 |
| M$^2$IXT (OFA$_{\text{BASE}}$) | 226M | 86.7 | 80.4 | 81.4 |
| M$^2$IXT (OFA$_{\text{LARGE}}$) | 528M | 90.1 | 85.0 | 85.9 |

Table 10: Ablation: Different-shots pretrained M$^2$IXT (OFA$_{\text{BASE}}$) evaluated with 1∼4 shots on COCO Caption.

| Pretrained\Evaluated (CIDEr) | 1-shot | 2-shot | 3-shot | 4-shot |
|---|---|---|---|---|
| 1-shot | 108.0 | 82.5 | 71.6 | 62.3 |
| 2-shot | 117.8 | 114.9 | 102.2 | 95.4 |
| 3-shot | 109.8 | 100.7 | 115.6 | 94.9 |
| 4-shot | 125.3 | 127.1 | 127.2 | 126.4 |

Table 11: Ablation: Randomly sampled 1/2/3-shots pretrained M$^2$IXT (OFA$_{\text{BASE}}$) evaluated with 1∼3 shots on COCO Caption.

| Pretrained\Evaluated (CIDEr) | 1-shot | 2-shot | 3-shot |
|---|---|---|---|
| 1/2/3-shots | 118.9 | 117.1 | 118.9 |

Table 12: Training Overhead: evaluated on M$^2$IXT (OFA$_{\text{BASE}}$), the time indicates running seconds for one training sample per GPU.

| Experiment | Training time |
|---|---|
| OFA | 0.2 |
| M$^2$IXT 1-shot | 0.5 |
| M$^2$IXT 2-shots | 0.6 |
| M$^2$IXT 3-shots | 1.2 |

Table 13: Inference Overhead: evaluated on four types of M$^2$IXT 2-shots, the time indicates running seconds for one training sample per GPU.

| Experiment | Inference time |
|---|---|
| M$^2$IXT (OFA$_{\text{TINY}}$) | 0.102 |
| M$^2$IXT (OFA$_{\text{MEDIUM}}$) | 0.147 |
| M$^2$IXT (OFA$_{\text{BASE}}$) | 0.206 |
| M$^2$IXT (OFA$_{\text{LARGE}}$) | 0.409 |

is fixed as `"What does the image describe?"`. For VQA questions and visual grounding objects, we invite some users to put forward their interests about the image. As shown in Figure 9. The displayed tasks are all open-ended, the VQA questions vary from sensing the object's color (*e.g.*, `"What is the color of the furniture?"`), determining the object's position (*e.g.*, `"Is the orange on the left side of the apple?"`), to asking what are the objects doing (*e.g.*, `"What are the sheep doing?"`), and counting the numbers of objects (*e.g.*, `"How many elephants are in the image?"`). The visual grounding objects are also quite di-

verse, like fruits (*e.g.*, apple, banana), furniture (*e.g.*, sofa), sign, animals (*e.g.*, sheep, baby elephant). For in-context examples, we select COCO caption (Chen et al., 2015) `val`, VQAv2 (Antol et al., 2015) `val` and visual grounding RefCOCO (Yu et al., 2016) `val` as the support sample dataset, which can be accessed publicly.

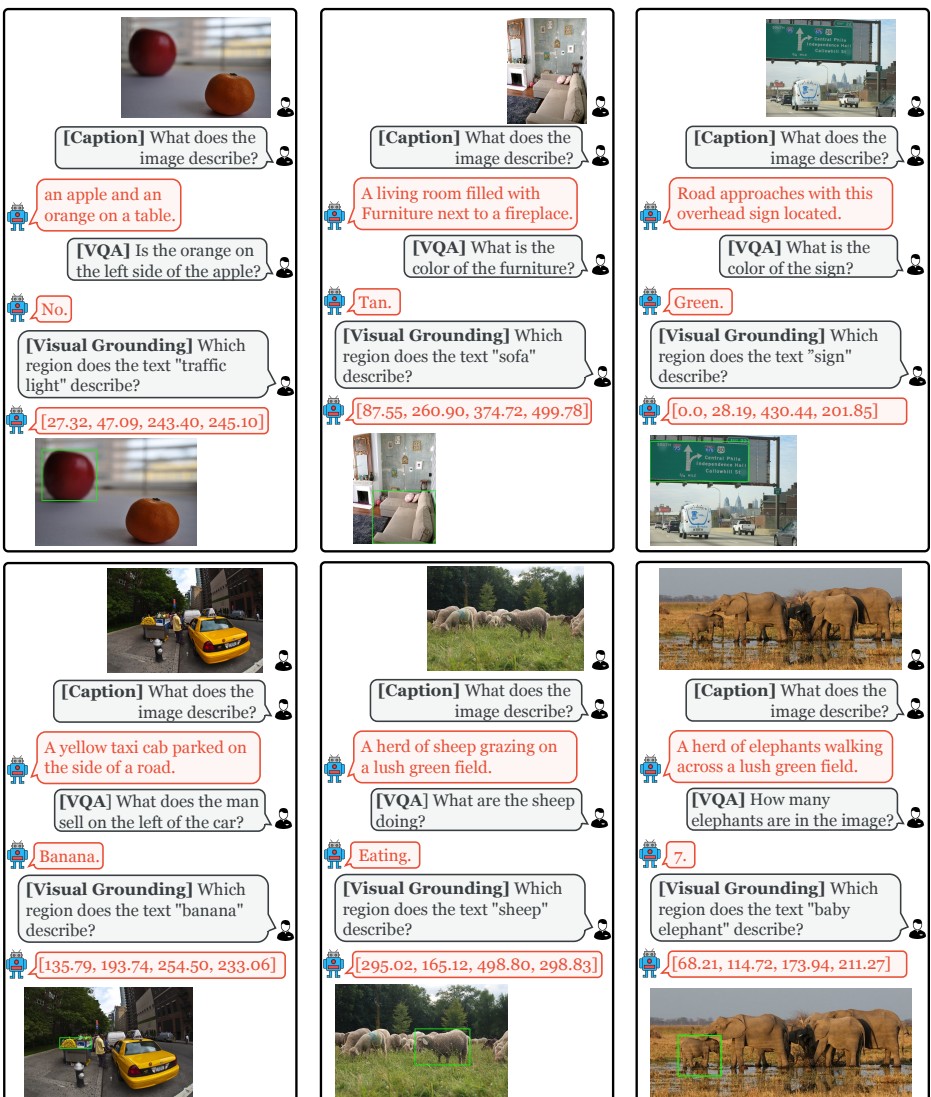

Figure 9: More **open-ended** M²IXT cases. The VQA questions and visual grounding texts are entered by invited users. The in-context examples are all from public dataset, and they are not shown in this figure.