# OpenReview forum: "Lightweight In-Context Tuning for Multimodal Unified Models"
_ICLR.cc/2024/Conference — ICLR 2024 Conference Withdrawn Submission_

### Official Review · Reviewer_1Zj1 · 2023-10-23

**Soundness:** 2 fair
**Presentation:** 3 good
**Contribution:** 2 fair
**Rating:** 5
**Confidence:** 4

**Summary:**

The paper introduces a method called MultiModal In-conteXt Tuning (M2IXT) to enhance the in-context learning (ICL) capabilities of multimodal unified models. ICL involves reasoning from contextual examples provided in multiple modalities like text, images, and coordinates. As more modalities are added, the understanding process becomes more challenging. Multimodal models often struggle to extrapolate effectively from contextual examples for ICL. M2IXT is a lightweight module that can be added to various multimodal unified models of different architectures and trained using a mixed-tasks strategy. It allows for the incorporation of various labeled examples from multiple modalities within an expandable context window. In conclusion, the proposed M2IXT is a simple and straightforward method that incorporates some references during training to improve the in-context learning capability of traditional multimodal models such as Flamingo.

**Strengths:**

Contribution:

According to the author's statement, the main contribution of this paper is to enhance the contextual capabilities of existing multi-modal models via continuing to train the model by introducing in-context examples. This method is not novel, due to similar method has been explored in previous method “MMICL: Empowering Vision-language Model with Multi-Modal In-Context Learning”. In addition, in NLP area, there are also many works to introduce in-context examples to improve the overall in-context learning capability of LLMs, such as “Learning to Retrieve In-Context Examples for Large Language Models”. In conclusion, I think the proposed method is not enough novel compared to previous works.

Presentation:

The writing style is clear. However, Figure 2 only simply describes the whole work process of model, leading to the training process unclear, especially for the target embedding network. The structure and function of this network are not clearly described in the method part of the paper.

Experiments:

Authors introduce many types of downstream tasks to verify their claims. Overall evaluation aspect is enough. However, I observe that lacking evaluation scores of some important and widely-used comparing models, such as BLIP-2 shown in Table 4. BLIP-2 are open-sourced and is a strong multimodal foundation model.

**Weaknesses:**

Contributions and Methods:

Multimodal In-Context Tuning: The paper builds upon the concept of multimodal in-context tuning as previously explored in "MMICL." In comparison to prior works, the authors introduce additional trainable parameters within multimodal language models, which include embedding tables and a target embedding table. However, the rationale behind incorporating these specific modules is not adequately explained. Particularly, the method section lacks clarity in describing the purpose and functioning of the target embedding table, leaving room for confusion. It is essential for the authors to provide a more comprehensive and explicit explanation of the role and significance of these added components.
Attribution of Improvement: The paper suggests that the enhanced performance is primarily attributed to the introduction of more in-context tuning data. While this is a valid point, it would be beneficial for the authors to elaborate on how the additional training data specifically contributes to the observed improvement. Providing insights into the mechanism through which the tuning data enhances the model's performance would enhance the clarity and depth of the paper's contributions.

Experiments:

Incorporation of Training Dataset: The paper achieves better performance compared to MMICL in Table 3, particularly in VQAv2. However, it's worth noting that the authors introduce the training dataset of VQA during their training process. Recent research, such as LLaVA-1.5, highlights the significance of multimodal data proposition for model performance. The authors should provide more clarity on the influence of this training data source on their results.
Missing Experimental Results: There are several missing experimental results in Table 4, which are crucial for drawing conclusions about the overall performance of the proposed method. The reliance on Flick30k alone is insufficient to establish the superiority of the author's method over previous MMICL or other multi-modal large oracle models. A more comprehensive set of experiments is needed.
Limited Models Comparison on RefCOCO (Table 2): The paper should incorporate and compare with a wider range of models on the RefCOCO dataset (Table 2). Expanding the comparison would provide a more comprehensive evaluation of the proposed method's performance.

Comparison with Existing Models: The paper should consider comparing its method with other models like FewVLM and Img2LLM, which have demonstrated good in-context learning capabilities. This comparison would help position the proposed method within the existing landscape of multimodal in-context learning approaches.
Additionally, the visual representation in Figure 1(a) showcases several multimodal models (Kosmos-1, Otter) with multimodal in-context learning capabilities. However, the experimental analysis in the paper does not incorporate these models. It would be beneficial for the authors to include a comparative analysis with these existing multimodal models to provide a more comprehensive assessment of their proposed approach.

**Questions:**

See the weakness part.

---

### Official Review · Reviewer_bTML · 2023-10-30

**Soundness:** 3 good
**Presentation:** 3 good
**Contribution:** 3 good
**Rating:** 6
**Confidence:** 3

**Summary:**

The paper proposes MultiModal In-conteXt Tuning (M2IXT) for in-context learning of multimodal unified models. The M2IXT module takes the contextual examples as input and outputs a sequence of token embeddings which can be concatenated with the query sequence embeddings. Experiments show M2IXT can significantly boost the few-shot ICL performance.

**Strengths:**

1. The paper is well-written and nicely-structured.
2. The proposed can be prepended to various multimodal unified models of different architectures and is easy to train.
3. Experiments show M2IXT can  significantly boost the few-shot ICL performance.

**Weaknesses:**

Please see the question part below

**Questions:**

1. In Table 1 and 2, the in-context samples are from the pre-training dataset. Why can these samples still provide information / additional knowledge to the pre-trained model to improve the evaluation metric? Did the in-context training samples and the proposed module provide a better input embedding layer? How well do these in-context samples perform on the pre-trained model?
2. Are in-context samples related to the query? (e.g. similar content or similar task?) How do the general in-context samples help / provide  additional knowledge for the query?
3. From Figure 7, it seems the tasks cannot generalize. Looks like performance improvement of the tasks is caused by including corresponding contexts during training. Are there methods to generalize across tasks?

---

### Official Review · Reviewer_kbgq · 2023-11-03

**Soundness:** 3 good
**Presentation:** 2 fair
**Contribution:** 3 good
**Rating:** 5
**Confidence:** 4

**Summary:**

The paper addresses how to augment existing multimodal unified models with an additional module enabling multimodal in-context learning. The M2IXT module, comprising three tunable components each of which processes the input image, instruction and target sequence, respectively, is prepended to a multimodal unified model and take as additional inputs a few contextual examples. Trained on a small subset of pretraining data, the M2IXT outperforms previous, more larger-scale, unified-models on a wide range of tasks with the in-context learning.

**Strengths:**

+ I agree with that the capability of in-context learning (ICL) is very important for multimodal foundation models because the ultimate goal of such approaches is making AGI, and ICL makes it eaiser to infer on unseen tasks and data, as shown in Section 4.3. From this point of view, I think the effectiveness of the proposed method endowing multimodal unified models with the ICL ability is quite high.
+ With a few training data and a lightweight additional module, the proposed method achieves performance gains on a wide range of tasks over baselines.

**Weaknesses:**

I cannot give this paper a high rating for the following reasons.
- The writing and presentation should be improved. It was quite hard to follow in several places when I first read the paper. Especially, the authors should revise the approach section (Section 3) for describing the architecture of M2IXT and training procedure in more detail. Figure 2 does not contain the details of the module.
- I think that one of the biggest advantages of ICL is enabling few-show adaptation to unseen tasks. However, most of the experiments conducted focus on evaluation on the tasks the model has seen during training, except Section 4.3. Additional experiments on unseen tasks will further demonstrate the effectiveness of the proposed method.

**Questions:**

- The authors said in A.1.2 that they created samples following OFA, e.g., transforming the caption sample into the VQA sample. It would be great to elaborate on this. Please explain more about this process.

---

### Official Review · Reviewer_rfnh · 2023-11-03

**Soundness:** 2 fair
**Presentation:** 3 good
**Contribution:** 2 fair
**Rating:** 3
**Confidence:** 4

**Summary:**

M2IXT is lightweight multimodal module for empowering (from 0 to 1) and enhancing few-shot learning ability of mutlimodal models.

**Strengths:**

1. M2IXT is a general plug-in module that can be applied to any multimodal models.
2. Experiments validate the effectiveness of M2IXT to improve the few-shot learning capabilities of existing multimodal unified models.

**Weaknesses:**

Let me first roughly define two types/eras of (multimodal) models.
1) Finetuning-style pretrained models (old-era), like BERT, BEiT-3, OFA. They show great finetuning benchmark performance. Some of them rely more on academicly curated labeled datasets for training, which could contribute to their benchmark performance. But they are not scalable. For these models, few-shot ability are typically not the focus.
2) Large models (new-era), like GPT-3, Flamingo. For these models, we mainly focus on their zero-shot / few-shot ability to perform a diverse range of tasks without finetuning. They are scalable and trained on large-scale web-based data, with a higher upper limit of capability.


You design your experiments on OFA. Models from the last era undoubtly have **no few-shot ability**, which is also not our focus. We typically care few-shot ability of large scalable models with higher capability upper limits, like GPT-3. Thus, you design is a bit weird and does not match the current trend.
This design also makes your work weak from two aspects:

**1. Make experiments weak.**

You claim your method is general to any model, but your experiments are almost limited to OFA/Unival and you main few-shot baseline is merely OFA. OFA has no few-shot ability because at that era, nobody cares few-shot for these models. Thus, it is obvious and trivial that training such models on in-context data can largely boost their few-shot performance.

In other words, I believe **for a model with no few-shot ability at all, any kind of design coupled with in-context data for training can easily empower it with better in-context ability, compared to the raw model with *no* in-context ability.** Thus, current experiments cannot support the effectiveness of your method and you claim.

**2. This practice is a bit meaningless from my perspective.**

If you care about the few-shot ability, you should position your focus on new-era large models, such as OpenFlamingo, Kosmos-1, Emu, etc. Small finetuning-style pretrained models, as its upper limit is bounded, are not worth exploring their few-shot ability.

**Inadequate and outdated literature review:**

Your focus is multimodal few-shot ability, but recent emerging multimodal fondation models with few-shot in-context learning ability are rarely mentioned and discussed. To name a few, Kosmos-1 [1], Emu [2].

[1] Language is not all you need

[2] Generative pretraining in multimodality

**Questions:**

Experiments on new-era large models can help, which position the focus of few-shot / in-context learning ability to the right place.